# Complex Nanowrinkling in Chiral Liquid Crystal Surfaces: From Shaping Mechanisms to Geometric Statistics

**DOI:** 10.3390/nano12091555

**Published:** 2022-05-04

**Authors:** Ziheng Wang, Phillip Servio, Alejandro D. Rey

**Affiliations:** Department of Chemical Engineering, McGill University, 3610 University St., Montréal, QC H3A 0C5, Canada; ziheng.wang@mail.mcgill.ca (Z.W.); phillip.servio@mcgill.ca (P.S.)

**Keywords:** cholesteric liquid crystal, anchoring, surface wrinkling, surface roughness

## Abstract

Surface wrinkling is closely linked to a significant number of surface functionalities such as wetting, structural colour, tribology, frictions, biological growth and more. Given its ubiquity in nature’s surfaces and that most material formation processes are driven by self-assembly and self-organization and many are formed by fibrous composites or analogues of liquid crystals, in this work, we extend our previous theory and modeling work on in silico biomimicking nanowrinkling using chiral liquid crystal surface physics by including higher-order anisotropic surface tension nonlinearities. The modeling is based on a compact liquid crystal shape equation containing anisotropic capillary pressures, whose solution predicts a superposition of uniaxial, equibiaxial and biaxial egg carton surfaces with amplitudes dictated by material anchoring energy parameters and by the symmetry of the liquid crystal orientation field. The numerical solutions are validated by analytical solutions. The blending and interaction of egg carton surfaces create surface reliefs whose amplitudes depend on the highest nonlinearity and whose morphology depends on the anchoring coefficient ratio. Targeting specific wrinkling patterns is realized by selecting trajectories on an appropriate parametric space. Finally, given its importance in surface functionalities and applications, the geometric statistics of the patterns up to the fourth order are characterized and connected to the parametric anchoring energy space. We show how to minimize and/or maximize skewness and kurtosis by specific changes in the surface energy anisotropy. Taken together, this paper presents a theory and simulation platform for the design of nano-wrinkled surfaces with targeted surface roughness metrics generated by internal capillary pressures, of interest in the development of biomimetic multifunctional surfaces.

## 1. Introduction

Biological cholesteric liquid crystals are organic materials [1] found in human compact bone (collagen) [2,3], beetles’ shells (chitin fibre) [4,5,6], plywood (cellulose) [7,8], virus RNA [9,10] and others [11]. Biological cholesteric liquid crystals have been found as a promising material in optics [12], tribology [13,14] and wetting applications [15,16]. The manifestation of molecular or colloidal orientation, chirality, self-assembly in nature’s liquid crystals [17,18,19,20,21,22] has served as a biomimetic model, and reciprocally, liquid crystal material physics has been used to shed light on biological structures [18,23,24,25,26,27,28,29]. Liquid crystals are orientationally ordered materials that flow like viscous liquids but display anisotropy like crystals [23] and usually form from high geometric aspect ratio molecular, supramolecular, and colloidal units, such as fibre, fibrils, and rods. The energetic interaction between the surface geometry and the anisotropy liquid crystal direction generates surface forces [20,30,31,32] and novel modes of intrinsic surface pattern formation explored in this paper.

The alignment of rod-like molecules determines the symmetry of liquid crystal phases. The two most common phases are the nematic liquid crystal (NLC) and cholesteric liquid crystal (CLC). In the NLC phase (Figure 1a), all the molecules tend to align toward a certain symmetry breaking direction, known as the director field or anisotropy axis. However, in the chiral CLC phase, the molecules have a layered structure. The preferred orientation within each layer in Figure 1b has a twisting angle with respect to its adjacent layer. The helix pitch $P_{0}$ corresponds to the length required to perform a full 360-degree rotation. Therefore, the director field in NCL can be regarded as a constant, while the director field in CLC is a periodic function with respect to its position.

The cholesteric phase is a preferred hierarchical structure adopted in many biological materials since it integrates mechanical, optical, tribological, and wetting functionalities. The blue colour observed on *Morpho* butterfly wings is the consequence of the presence of a chiral structure in the chitin fibres [33]. The beetles’ shells, on the other hand, generate a unique structural colour that changes with different polarizer orientation [34]. The chitin fibres are stacked in a chiral structure, therefore showing selectivity to light [5]. Cholesteric chitin fibres can enhance the mechanical properties of some animals’ organs as well. For example, the dactyl clubs of Odontodactylus scyllarus mantis shrimp can accelerate the dactyl heel of 65 to 104 km/$s^{2}$ and can reach a peak speed of 23 m/$s^{-1}$ from a stationary position [35,36]. A double-spiral defect pattern is observed in the chitin fibres since it creates a larger surface area that can easily dissipate the total energy when the fracture propagates. The multifunctional cholesteric liquid crystal is therefore a promising biomimetic material that can be applied in many fields.

The self-assembly and self-organization of cholesteric liquid crystal with free surfaces or interfaces usually generates complex multi-directional and multi-wavelength periodic wrinkling patterns [37,38,39,40,41]. The origin of these wrinkles is the presence of anisotropic surface/interfacial tension due to the anisotropic axis, its spatial gradients, and its chirality. These multiscale wrinkling surface reliefs with distinct morphology generate material functionalities in diffraction gratings [42,43,44,45] and elastomers [46]. However, a lack of a fundamental understanding of wrinkling mechanisms and geometry and the challenges in experimental characterization creates barriers to the biomimetic development of liquid crystalline wrinkled surfaces with controllable and targeted geometries. Thus, the opportunity of using liquid crystal material science and self-assembly coupled with computational geometry and high-performance computing offers a new tool in this functional surfaces’ area.

In our previous work [47] on 2D surface wrinkling, using the lowest possible order anisotropic surface energy, we captured the emergence of single and doubly periodic egg carton surfaces of the type:(1)$$\begin{array}{cc} h_{\mathrm{uniaxial}}\propto & \cos\left(\frac{2\pi x}{P_{0}}\right)\;\mathrm{or}\;\cos\left(\frac{2\pi y}{P_{0}}\right)\\ h_{\mathrm{equibiaxial}}\propto & \cos\left(\frac{2\pi x}{P_{0}}\right)\cos\left(\frac{2\pi y}{P_{0}}\right)\\ h_{\mathrm{biaxial}}\propto & \cos\left(\frac{2\pi nx}{P_{0}}\right)\cos\left(\frac{2\pi my}{P_{0}}\right),\;n\neq m=1,2,\cdots \end{array}$$

Furthermore, we showed [47] that in the linear small-amplitude regime, the superposition of these egg carton modes predicted by this simple model leads to surface reliefs rich in geometric diversities, such as minimal surface patches, umbilic points, zero Gaussian curvature patches and more. Encouraged by this previous work, here we extend this previous work on egg carton surfaces considering higher-order anisotropic surface energy models. Two important novel features of surface pattern formation in liquid crystal surfaces, exploited in this paper, are: 1The dimensionless anisotropic-to-isotropic interfacial tension anchoring coefficient ratio (*r*) transforms orientation features into geometric surface features [48]. If *r* is relatively small, material orientation scales of the order microns generate surface nanowrinkling, otherwise submicron or micron features will arise;2The spatial orientation of the director field n(x) usually depends on current or prior self-assembly [49] or self-organization conditions [50] and hence this key input to the surface shape equation can generate surface wrinkling complexity not obtained by external stress loading of elastic surfaces.

Complex wrinkling patterns can be formed by various mechanisms and predicted by various models. For example, the nonlinear membrane elastic model [51], the dynamic approach by using the Landau–Ginzburg dynamics equations [52], the anisotropic spontaneous curvature model [53], and surface wrinkling due to temperature changes [46] or plasma treatment [54]. However, all the approaches mentioned above require an external effect that is independent of the material itself, such as the presence of an external load. In the present paper, we present an intrinsic self-assembly approach where surface wrinkling is only developed by the anisotropic interfacial tension (anchoring effect) and no other surface force fields or external loads are necessary. In other words, the coupling of surface geometry and surface orientation is the geometric morphing agent. In particular, this research focuses on the lyotropic cholesteric liquid crystals inspired by biological systems where the temperature is the ambient temperature. The concentration of the liquid. The typical concentration for this type of material is of the order of 50 mg ml ^−1^ [55,56,57], and the environment (such as humidity) could affect the surface profile [58]. The other type of liquid crystal is the thermotropic liquid crystal, whose properties are affected by temperature, including, for example, the wetting properties [59], anchoring and optics [60], as well as coatings and surface roughness [61].

Surface roughness plays a critical role in various biological and biomimetic systems. For example, a biomimetic approach inspired by shark skin improves aerodynamic performance [62]. Multiscale roughness-induced hydrophobicity can be found in lotus and butterfly wings [63], as well as osteogenic biomimetic surfaces [64]. The colours of butterfly wings are due to their multiscale surface patterns [65], and the surface roughness is later shown to determine the imaging quality in an AR/VR display system [66]. The usual characterization of surface roughness geometry is based on the distribution function of the surface relief height and its moments. The statistical properties of the surface geometry are described by its roughness (second-order moment), the skewness (third-order moment) and the kurtosis (fourth-order moment). The average roughness and root-mean-square roughness contains the mean value and its overall deviation from the mean, while the higher order skewness and kurtosis moments capture surface symmetry breaking features that can generate surface functionalities such as static friction [67] and biological surface topography [68]. For example, a surface with positive skewness has a much higher shear strength than a negative skewness given the same average roughness (first-order moment) and root-mean-square roughness (second-order moment) [69]. The surface kurtosis quantifies features such as isolated peaks and valleys. In this research (Section 3.3), we propose an integrated analytical approach to obtain wrinkled surface profiles with rich skewness and kurtosis features.

Based on the previous discussion, Figure 2 summarizes the key issues addressed in this paper: (a) the surface wrinkling pattern formation process is due to an instability of a flat surface above the orientation director field n(x) which is unstable and tends to relax to a wrinkled surface. In this paper, we address the mechanism using an efficient shape equation, a complex director orientation and a high order interfacial energy formulation by keeping the quartic term in the surface free energy in Equation (Equation 2). (b) The three primary liquid crystal orientation gradients are known as splay, twist and bend; the surface wrinkling process is generated by bend and splay modes whose presence, absence, and relative mixing impact the wrinkling geometry. In this paper, we address how the different modes impact final shapes explicitly, which is crucial in optimizing and controlling surface reliefs. (c) The wrinkled surface can be decomposed into various fundamental wrinkling modes (egg cartons, corrugated, etc.), whose contribution to the final wrinkling surface is captured by a revealing shape C-matrix (Equation (Equation 15)); the figure shows a 5×5 matrix with each element corresponding to wrinkling patterns. In this paper, we found optimal ways to synthesize information for the wrinkling patterns using a shape classification based on the superposition of primary modes. (d) Finally (not shown in Figure 2), an in-depth characterization of shape statistic and scaling is provided, aiming at future potential applications in tribology, friction, structural colour and more.

The scope of this paper is restricted to equilibrium patterns, and all time-dependent phenomena are not considered. The liquid crystalline phase is described by a director orientation field, and issues arising from molecular ordering are not considered. The director field is a known input to the overall model, and hence, there is no effect of geometry on material structure. The chosen defect-free director field is selected as it manifests a rich potential for wrinkling. The nonlinear (quartic) anisotropic interfacial energy is a well- known [70,71], widely used function of (n·k), where k is the surface unit normal. Tangential Marangoni forces of any origin are not considered. The essential nature of the model is the liquid crystal shape equation, and the sole focus is on intrinsic capillary pressures; hence, bulk elastic corrections and bulk stress jumps are not considered. Future work can include all these effects as required by experimental data, biological observations and material innovation. An effort is made to highlight mechanisms, novelty and significance, while mathematical details are found in the appendices. Without ambiguity, in this paper, we refer to the anisotropic interfacial tension coefficient as the anchoring coefficient and anisotropic interfacial energy as the anchoring energy.

The organization of this paper is as follows. In Section 2.1, we present a summary of the surface energy expression and the liquid crystal director elastic deformation. The governing equation of surface wrinkling and the mechanism are explained in Section 2.2, with a linear approximation proposed in Section 2.3. The results are presented in Section 3, where we compute the morphology of a wrinkling surface (Section 3.2) and the skewness-kurtosis profiles (Section 3.3). In Section 4, we characterize surface roughness parameters in the surface anchoring (anisotropic interfacial energy) parametric space. Section 5 summarizes key results and concludes with future work. Appendix A presents the fundamental equations that are used to describe the intrinsic geometry, and Appendix B provides details on the wrinkling scaling laws (Equation (Equation 19)).

## 2. Theory and Model Formulation

### 2.1. Surface Energy and Elastic Deformation

The anisotropic surface free energy of liquid crystal is described by the generalized Rapini–Papoular equation [20,70,71]
(2)$$\gamma =\gamma_{0}+\sum_{j=1}^{m}\mu_{2j}{\left(n\;\cdot \;k\right)}^{2j}$$
where $\gamma_{0}$ is the isotropic surface tension, and $\mu_{2j}\left(j=1,2,\cdots \right)$ are the temperature- and/or chemical composition-dependent anchoring coefficients. The anisotropy is presented in the product of the material director field n and geometric surface unit normal k. In this paper, we focus on the quartic model such that $\mu_{4}\neq 0$ and $\mu_{6}=\mu_{8}=\cdots =0$. The dimensionless quartic model used in this paper reduces to $\gamma^{*}=1+\epsilon_{2}{\left(n\;\cdot \;k\right)}^{2}+\epsilon_{4}{\left(n\;\cdot \;k\right)}^{4}$. The signs of the material parameters $\epsilon_{j}\left(j=1,2\right)$ are restricted as follows. The thermodynamic stability criterion $\gamma^{*}>0$ within the region $0\leq {\left(n\;\cdot \;k\right)}^{2}\leq 1$ requires $\gamma^{*}>0$. This criterion implies that not every combination of $\epsilon_{2}$ and $\epsilon_{4}$ has physical meaning. The purple region in Figure 3 represents the possible choices in the anchoring parametric space $\left(\epsilon_{2},\epsilon_{4}\right)$ by assuming that $\gamma^{*}>0$ for every $0\leq {\left(n\;\cdot \;k\right)}^{2}\leq 1$. If $\epsilon_{4}=0$, Equation (Equation 2) degenerates to the quadratic model, which is discussed in [58,72,73,74,75], and it is not included or discussed here.

The stable regions in Figure 3 are separated into 8 different sub-regions based on the location of the local maximum and minimum of the surface energy density γ. The critical straight lines in Figure 3 are $L_{1}:\epsilon_{4}=-\epsilon_{2}$, $L_{2}:\epsilon_{4}=-\epsilon_{2}-1$ and $L_{3}:\epsilon_{4}=-\epsilon_{2}/2$. The critical parabolic curve $L_{4}$ is $\epsilon_{4}=\epsilon_{2}^{2}/4$. Any region that is outside the purple part in Figure 1 does not ensure that $\gamma^{*}>0$ holds for every independent director n and unit normal k. Therefore, those regions are thermodynamically unstable. For nanowrinkling where both $\epsilon_{2}$ and $\epsilon_{4}$ are on the order of ≪$10^{0}$, surface tension is always positive, and the stable condition is always satisfied.

Once an anchoring coefficient pair $\left(\epsilon_{2},\epsilon_{4}\right)$ is chosen, the surface energy density profile $\gamma^{*}$ is uniquely determined and is a function of ${\left(n\;\cdot \;k\right)}^{2}$, where n and −n is equivalent to the system in the framework of the non-polar liquid crystal model. The local minimum or maximum of $\gamma^{*}$ is only determined by the anchoring coefficient ratio $r=\epsilon_{2}/\left(2\epsilon_{4}\right)$. The position (in terms of (n·k)) of the local minimum and maximum of all 8 stable regions separated by 4 critical curves $L_{1}$ to $L_{4}$ in Figure 3, A, B1, B2, C, D, E1, E2 and F, are summarized in Table 1.

Table 1 demonstrates that the local minimum or maximum of the surface energy density $\gamma^{*}$ only occurs either at n‖k, n⊥k, or ${\left(n\;\cdot \;k\right)}^{2}=\left|r\right|$, corresponding to planar (tangential director to the surface) anchoring, homeotropic (perpendicular director to the surface) anchoring and oblique (tilted director to the surface) anchoring, respectively. The local energy density $\gamma^{*}$ under these three conditions are $1+\epsilon_{2}+\epsilon_{4}$, 1 and $1+\epsilon_{2}\left|r\right|+\epsilon_{4}r^{2}$, respectively. The significance of Table 1 is that it shows the parametric bubble where a spatially variable director field samples the energy extrema and the types of extrema. For example, in region B1, one can see that if n‖k in one region adjacent to ${\left(n\;\cdot \;k\right)}^{2}=\left|r\right|$, the energy landscape will drive a geometric change through surface tilting, as shown below.

For an isotropic surface tension, the only capillary pressure is the Laplace pressure generated by the mean curvature or divergence of k. For a liquid crystal, we have other sources of capillary pressures [20] generated by the director n gradients, captured by the splay S, twist T and bend B deformation given by [76,77]
(3)S=∇·n,T=n·∇×n,B=n×∇×n

In partial summary, the parametric anchoring coefficient space (Figure 3a), together with the surface energy (Equation (Equation 2)) and the director field and its deformation modes (Equation (Equation 3)), are the primary information model needed to analyze nanowrinkling.

### 2.2. Cahn–Hoffman Capillary Vector Method

The surface shape equation is the interfacial normal stress balance equation. This balance equation is given by the sum of all the capillary forces and the bulk normal stress jump (the difference between normal stresses in the two bulk phases) [20]; see [20] for details.

The Cahn–Hoffman capillary vector ξ is defined by [78,79,80]
(4)$$\xi =\xi_{\|}+\xi_{⊥},\;\xi_{\|}=I_{\left(\sigma \right)}\;\cdot \;\frac{\partial \gamma }{\partial k},\;\xi_{⊥}=\gamma k$$
where $I_{\left(\sigma \right)}=I-\mathrm{kk}$ is the surface projection tensor. The physical interpretation of Equation (Equation 4) is demonstrated with Figure 4; for simplicity, we only consider a quadratic anchoring model. In Figure 4a,b, the tangential component of the Cahn–Hoffman capillary vector $-\xi_{\|}$ gives the direction where the rotation of the surface unit normal k around $b_{0}$ results in the fastest rate of decreasing anchoring energy.

In the absence of external loads and zero bulk stress jumps, the governing shape implies a surface divergence free ξ, which reveals three fundamental capillary pressures by applying the chain rule in Equation (Equation 4): (1) the change in the local surface area (variation of k along normal direction) causes dilation pressure; (2) the change in the local surface orientation (variation of k along tangent direction) causes rotation pressure, unique to anisotropic media; (3) the director pressure induced by surface gradients in the average molecular orientation [80].
(5)$$-\nabla_{\left(\sigma \right)}\;\cdot \;\xi =\underset{\mathrm{Dilation Pressue}}{\underbrace{-\frac{\partial \xi_{⊥}}{\partial k}:{\left(\nabla_{\left(\sigma \right)}k\right)}^{T}}}\;\underset{\mathrm{Rotation Pressue}}{\underbrace{-\frac{\partial \xi_{\|}}{\partial k}:{\left(\nabla_{\left(\sigma \right)}k\right)}^{T}}}\;\underset{\mathrm{Director Pressue}}{\underbrace{-\frac{\partial \xi_{\|}}{\partial n}:{\left(\nabla_{\left(\sigma \right)}n\right)}^{T}}}\;=0$$

Replacing γ from Equation (Equation 2) to Equation (Equation 4) yields the three dimensionless fundamental pressures (scaled by $\gamma_{0}$) in Equation (Equation 5)
(6)$$\begin{array}{cc} \mathrm{Dilation}\;{\mathrm{Pressure}}^{*}= & -2H\left(1+\sum_{j=1}^{m}\epsilon_{2j}{\left(n\;\cdot \;k\right)}^{2j}\right) \end{array}$$
(7)$$\begin{array}{cc} \mathrm{Rotation}\;{\mathrm{Pressure}}^{*}= & \sum_{j=1}^{m}2j\epsilon_{2j}{\left(n\;\cdot \;k\right)}^{2j-2}\left(\left(2j-1\right)\mathrm{nn}:\nabla_{\left(\sigma \right)}k+2{\left(n\;\cdot \;k\right)}^{2}H\right) \end{array}$$
(8)$$\begin{array}{cc} \mathrm{Director}\;{\mathrm{Pressure}}^{*}= & \sum_{j=1}^{m}2j\epsilon_{2j}{\left(n\;\cdot \;k\right)}^{2j-2}\left(\left(2j-1\right)\mathrm{kn}:\nabla_{\left(\sigma \right)}n+\left(n\;\cdot \;k\right)\mathrm{tr}\left(\nabla_{\left(\sigma \right)}n\right)\right) \end{array}$$
where $H=-\nabla_{\left(\sigma \right)}\;\cdot \;k$ is the mean curvature. The cancellation of the final net pressure (Equation (Equation 5)) indicates that the wrinkling phenomenon arises due to the balance of the three capillary pressures. Equations (Equation 6) to (Equation 8) show the additional capillary pressures from geometry-orientation couplings such as: n·k, $\mathrm{nn}:\nabla_{\left(\sigma \right)}k$, $\mathrm{kn}:\nabla_{\left(\sigma \right)}n$.

### 2.3. Linear Approximation

It turns out that given the complexity of the shape in Equation (Equation 5), the linear regime is an indispensable tool to characterize, analyze, and shed light on the underlying principles of pattern formation. In the linear region ($|\epsilon_{2}|\ll 1$ and $|\epsilon_{4}|\ll 1$), the rotation pressure is relatively small compared to the dilation pressure and director pressure. The following identities hold: (9)$$\begin{array}{c} \mathrm{kn}:\nabla_{\left(\sigma \right)}n\approx -\left(\frac{1}{2}\nabla {\left|n\right|}^{2}-\left(n\;\cdot \;\nabla \right)n\right)\;\cdot \;{\hat{\delta }}_{z}=-B\;\cdot \;{\hat{\delta }}_{z} \end{array}$$(10)$$\begin{array}{c} \left(n\;\cdot \;k\right)\mathrm{tr}\left(\nabla_{\left(\sigma \right)}n\right)\approx \left(n\;\cdot \;{\hat{\delta }}_{z}\right)\nabla \;\cdot \;n=S\left(n\;\cdot \;{\hat{\delta }}_{z}\right) \end{array}$$

Equation (Equation 9) and Equation (Equation 10) demonstrate that in the director pressure (Equation (Equation 8)), the deformation is only presented by the bending elastic mode and splay elastic mode (see Equation (Equation 3)). A further approximation yields the Laplacian $\nabla^{*2}h^{*}$ for the surface relief:(11)$$\nabla^{*2}h^{*}=\sum_{j}^{m}2j\epsilon_{2j}\left(\frac{\partial }{\partial x^{*}}\left({\left(n_{z^{*}}\right)}^{2j-1}n_{x^{*}}\right)+\frac{\partial }{\partial y^{*}}\left({\left(n_{z^{*}}\right)}^{2j-1}n_{y^{*}}\right)\right)$$
where $n={\left[\begin{array}{ccc} n_{x^{*}} & n_{y^{*}} & n_{z^{*}} \end{array}\right]}^{T}$ is expanded in the orthonormal basis $\left\{\delta_{x^{*}},\delta_{y^{*}},\delta_{z^{*}}\right\}$, and the dimensionless variables are
(12)$$\nabla^{*}=P_{0}\nabla ,\;x^{*}=\frac{x}{P_{0}},\;y^{*}=\frac{y}{P_{0}},\;z^{*}=\frac{z}{P_{0}},\;h^{*}=\frac{h}{P_{0}}$$

This generic result shows the direct connections between the surface geometry and director splay/bend deformations. It can be seen that increasing the surface energy nonlinearity “*m*” with a fixed director field increases the number of shaping modes. In addition, the symmetry properties of the terms in Equation (Equation 11) shows that the normal director orientation ($n_{z^{*}}$) plays a unique role in generating curvature.

The shape Equation (Equation 11) provides the groundwork for a new way to classify and extract surface wrinkling information which would not be possible with numerical simulations alone. In the linear region, the surface profile $h^{*}\left(x^{*},y^{*}\right)$ can be analytically solved using Equation (Equation 11). The resulting $h^{*}$ is a linear combination of fundamental wrinkling modes $h^{*}=w_{x^{*}}^{T}\;\cdot \;Q\;\cdot \;w_{y^{*}}$ where a director field n with a doubly helical structure and an *m*-th order surface energy (Equation (Equation 2)) generates the following geometric m×m matrix Q and basis shape vectors $w_{i}\left(i=x^{*},y^{*}\right)$:(13)$$Q_{m}=\left[\begin{array}{cccc} \cdots & \leftarrow & \omega_{y^{*}} & \to \\ ↑ & \ddots & \vdots & \vdots \\ \omega_{x^{*}} & \cdots & \ddots & \vdots \\ ↓ & \cdots & \cdots & \ddots \end{array}\right],\;w_{x^{*}}=\left[\begin{array}{c} \cos0\\ \cos2\pi x^{*}\\ \cdots \\ \cos2\pi \left(2m-1\right)x^{*}\\ \cos2\pi 2mx^{*} \end{array}\right],\;w_{y^{*}}=\left[\begin{array}{c} \cos0\\ \cos2\pi y^{*}\\ \cdots \\ \cos2\pi \left(2m-1\right)y^{*}\\ \cos2\pi 2my^{*} \end{array}\right]$$

The entries of the generic Q-matrix are $\epsilon_{i}$-dependent (i=1,⋯,m) constants whose values reflect the symmetry and periodicities of the director field and the anchoring model adopted (Equation (Equation 2)), as expected from our discussions above on geometry-orientation couplings. For a quartic anchoring model (m=2), the Q-matrix is 5×5, indicating 24 fundamental modes, excluding the trivial flat mode. If a director $n_{0}$ of the following form is given, we can solve for the Q-matrix, resulting in a shape C-matrix for the given director field $n_{0}$
(14)$$n=\left[\begin{array}{c} \sin2\pi x^{*}\cos2\pi y^{*}\\ \sin2\pi y^{*}\\ \cos2\pi x^{*}\cos2\pi y^{*} \end{array}\right],\;Q\left(\epsilon_{2},\epsilon_{4}\right)\overset{n_{0}}{\to }C\left(\epsilon_{2},\epsilon_{4}\right)$$

The Q-matrix contains 24 fundamental wrinkling modes, and the C-matrix reduces the entries to 10 nonzero modes due to the symmetry present in the director $n_{0}$.

Next, we present a specific application based on Equations (Equation 11) and (Equation 13). The wrinkling mode $\cos2\pi \omega_{x^{*}}x^{*}\cos2\pi \omega_{y^{*}}y^{*}$ is denoted as $\left[\omega_{x^{*}},\omega_{y^{*}}\right]$ for clarity. As mentioned above, we use a quartic model m=2 (Equation (Equation 2)) with two-dimensional parametric space $\left(\epsilon_{2},\epsilon_{4}\right)$ (Figure 3). For the egg carton-type surfaces of interest in this work, we introduce the following compact nomenclature (briefly mentioned in the introduction): (1)uniaxial egg carton modes: $\left[0,\omega_{y^{*}}\right]$ or $\left[\omega_{x^{*}},0\right]$;(2)equibiaxial egg carton modes: $\left[\omega_{x^{*}},\omega_{y^{*}}\right]$ and $\omega_{x^{*}}=\omega_{y^{*}}$;(3)biaxial egg carton modes: $\left[\omega_{x^{*}},\omega_{y^{*}}\right]$ and $\omega_{x^{*}}\neq \omega_{y^{*}}$. 

Hence, (1) denotes a 1D surface corrugation, (2) denotes the ideal equi-doubly periodic egg carton, and (3) denotes an egg carton with different *x*- and *y*-wavelengths, as follows. Assuming $\epsilon_{4}\neq 0$ (for $\epsilon_{4}=0$ please consult [47]), the analytic solution $h^{*}$ ($h^{*}=w_{x^{*}}^{T}\;\cdot \;C\left(\epsilon_{2},\epsilon_{4}\right)\;\cdot \;w_{y^{*}}$) to Equation (Equation 11) can be rearranged such that
(15)$$h^{*}=-\frac{2\epsilon_{4}}{\pi }w_{x^{*}}^{T}\;\cdot \;C\left(r\right)\;\cdot \;w_{y^{*}},\;C\left(r\right)=\left[\begin{array}{ccccc} 0 & 0 & 0 & 0 & 0\\ 0 & 0 & (3+8r)/40 & 0 & 3/136\\ (3+8r)/64 & 0 & (1+2r)/32 & 0 & 1/320\\ 0 & 0 & 1/104 & 0 & 1/200\\ 3/256 & 0 & 1/80 & 0 & 1/512 \end{array}\right]$$

Here, the shape matrix C is the specific realization of the generic Q-matrix (Equation (Equation 13)). The relationship between C(r) and $C(\epsilon_{2},\epsilon_{4})$ is $C\left(\epsilon_{2},\epsilon_{4}\right)=-\left(2\epsilon_{4}/\pi \right)C\left(r\right)$. It is important to note that we have expressed the coefficient matrix $C(\epsilon_{2},\epsilon_{4})$ in terms of an *r*-dependent coefficient matrix C(r) multiplied with the coefficient $\epsilon_{4}$. Equation (Equation 15) reveals that *r* only controls 3 modes: [1,2] (biaxial), [2,0] (uniaxial) and [2,2] (equibiaxial). The [1,2] and [2,0] modes vanish at the same time when r=−3/8, and the [2,2] mode does not contribute when r=−1/2. We can conclude that $\epsilon_{4}$ directly controls the surface profile magnitude, while *r* determines the surface morphology through the coefficient matrix C. The nullification of fundamental modes at specific negative *r*-values indicates the impact of the plus or minus effect and the importance of the II and IV quadrants in the parametric space (Figure 3) to generate shape transitions and extrema.

Figure 5 shows the representation of the wrinkling modes by the generic Q-matrix (a) and (b) the corresponding specific C-matrix. The Q-matrix is all of the theoretically possible 24 wrinkling modes, and the C-matrix, as mentioned above, is the specific coefficient matrix in this paper (determined by the director field n in Equation (Equation 13) and a quartic anchoring model $\left(\epsilon_{4},r\right)$). In our adopted model (C-matrix), Figure 5b shows that there are only 10 fundamental wrinkling modes: 2 uniaxial modes (cylindrical wrinkling), 2 equibiaxial modes (symmetric wrinkling) and 6 biaxial modes (asymmetric wrinkling). The rest of the wrinkling modes in the Q-matrix (indicated by black zones in (b)) do not contribute to the surface profile. Importantly, the anchoring coefficient ratio modulates the amplitude of the three modes (see Equation (Equation 15)).

In partial summary, symmetry analysis combined with basic liquid crystal surface physics can be systematically represented by an m×m shape matrix Q containing the fundamental wrinkling modes for an *m*-th anchoring model. The symmetry of the adopted director field then reduces the number of fundamental modes in the Q-matrix, and the reduced actual shape C-matrix eliminates forbidden modes. For the adopted $n_{0}$-vector, rich and complex wrinkling modes remain active. The systematic approach formulated here can efficiently be applied to any director field and any anchoring model, as required by future experimental observations and/or computational discovery.

## 3. Results and Discussion

### 3.1. Parametric Space and Numerical Methods

The governing Equation (Equation 5) is a second-order nonlinear partial differential equation of $h^{*}\left(x^{*},y^{*}\right)$ with respect to two variables $x^{*}$, $y^{*}$. We replace 0 with $dh^{*}/dt^{*}$ in the governing equation, and the numerical solution converges when $t^{*}\to \infty$. We use the EPS method [81,82,83] along the time dimension ($h_{\mathrm{new}}-h_{\mathrm{last}}+\Delta h$ until ∥Δh∥<δ) and the finite difference method on the two spatial directions with periodic boundary conditions (matrix index satisfies $h_{1a}=h_{Na}$ and $h_{a1}=h_{Na}$, where a=1,⋯,N). In this research, N=200, $\delta =dt=10^{-6}$.

*h* shows a translational invariance in Equation (Equation 5) such that if h→h+constant, k and $\nabla_{\left(\sigma \right)}$ does not change from Equation (Equation 6) to (Equation 8), which meets our expectation since lifting or lowering the surface should not affect the morphology. To obtain a consistent result in numerical simulation, the final surface profile is $h=h_{\mathrm{converged}}-\mathrm{mean}\left(h_{\mathrm{converged}}\right)$, which ensures mean(h)=0. The numerical results were compared with analytical results for the linear regime (see Section 2.3), and in all cases, we find an essentially perfect agreement.

### 3.2. Surface Profile

Equation (Equation 15) implies that $\epsilon_{4}$ only controls the absolute magnitude of the wrinkling profile, while the anchoring coefficient ratio *r* determines the morphology. Once $\epsilon_{4}$ is fixed, *r* can be chosen arbitrarily. Here, we focus mainly on |r|≤2 (see Figure 3). The numerical solution, analytic linear approximation (via solving Equation (Equation 11)) and their cross-sectional plots for r=−2, r=−0.258, r=−0.454, r=−0.416 and r=+2 are presented in Figure 6. These specific anchoring coefficient ratio values are chosen because each of them corresponding to a special case in the surface skewness and kurtosis analysis discussed later in Section 3.3. In the parametric space (Figure 3), the specific *r*-values correspond to: r=−2, zone F (slope=−0.25); r=−0.454, −0.416 and −0.258 are in zone E1 (slope<−1); and r=+2 is in zone D (slope=+0.25). The average difference ratio Ar is defined as
(16)$$\mathrm{Ar}=\frac{\sqrt{\mathrm{tr}(d\;\cdot \;d^{T})}}{N^{2}}\times 100\%,\;d=\frac{h_{\mathrm{numerical}}-h_{\mathrm{analytic}}}{(\max\left(h_{\mathrm{numerical}}\right)-\min\left(h_{\mathrm{numerical}}\right))/2}$$
where h is the solution matrix with N×N entries, and the division represents element-wise division.

Figure 6 presents the dimensionless surface wrinkling profile $h^{*}$ as a function of two spatial variables $\left(x^{*},y^{*}\right)$. The first row demonstrates the numerical solution, and the second row shows the analytic solution. The cross-section plots along the diagonal curve $y^{*}=x^{*}$ and the two boundaries $x^{*}=0$ and $y^{*}=0$ are shown in the third row of Figure 6. The full and dashed curves represent the numerical cross-section plot and analytic cross-section plot, respectively. The average difference ratio Ar in Figure 6 varies from 0.0114% to 0.0613%, validating the fact that the linear model fits the numerical solution accurately. The dimensionless surface profile $h^{*}$ has an order of magnitude of $10^{-3}$, implying a nanowrinkling profile. The results shown in Figure 6 demonstrates the complex wrinkling patterns that result from the superposition of fundamental egg carton shape modes given in Equations (Equation 13) and (Equation 15). Comparing Figure 6(a1,e1), we find that $h^{*}\left(r\right)\neq -h^{*}\left(-r\right)$. This relationship verifies that the C-matrix does not contain any symmetry with respect to r=0, unlike the relationship $h^{*}\left(r\right)=-h^{*}\left(-r\right)$ found in the 2D surface wrinkling phenomenon [84].

In partial summary, the model and simulation are able to capture non-ideal complex wrinkling patterns whose symmetries and periodicities can be modulated by the anchoring coefficients. In practical terms, these coefficients are functions of the chemistry, concentration, and temperature, offering a number of routes to fix the value of *r* and hence target different egg cartons in a more predictable way.

### 3.3. Skewness and Kurtosis

The statistics of the surface geometry given by moments of the surface relief are required to establish potential applications that depend on roughness. Here the superposition of various egg carton will be shown to generate a rich response to the anchoring coefficient ratio “*r*”. In what follows, we only consider the morphology factor *r* in the scaled dimensionless surface profile $\bar{h}=-\pi h^{*}/\left(2\epsilon_{4}\right)$ to eliminate the effect of $\epsilon_{4}$ on $h^{*}$, since it only linearly increases the magnitude of $h^{*}$.

The surface roughness parameters root mean square (${\mathrm{Sq}}^{2}$), skewness (Ssk) and kurtosis (Sku) are defined by (with a mean value of $\bar{h}=w_{x^{*}}^{T}\;\cdot \;C\;\cdot \;w_{y^{*}}$ equal to 0)
(17)$$\begin{array}{c} A={\;\int \int \;}_{A}\sqrt{\bar{g}}dx^{*}dy^{*},\;{\mathrm{Sq}}^{2}=\frac{1}{A}{\;\int \int \;}_{A}{\bar{h}}^{2}dx^{*}dy^{*}, \end{array}$$
(18)$$\begin{array}{c} \mathrm{Ssk}=\frac{1}{A{\mathrm{Sq}}^{3}}{\;\int \int \;}_{A}{\bar{h}}^{3}dx^{*}dy^{*},\;\mathrm{Sku}=\frac{1}{A{\mathrm{Sq}}^{4}}{\;\int \int \;}_{A}{\bar{h}}^{4}dx^{*}dy^{*} \end{array}$$
where $\bar{g}=1+{\left(\partial \bar{h}/\partial x^{*}\right)}^{2}+{\left(\partial \bar{h}/\partial y^{*}\right)}^{2}$ is the metric. If |r|≫1, the following scaling laws hold (see Appendix B)
(19)$$A\propto r,\;\mathrm{Sq}\propto \sqrt{r},\;\mathrm{Ssk}\propto \sqrt{r},\;\mathrm{Sku}\propto r,\;\mathrm{and}\;\mathrm{Sku}\propto {\mathrm{Ssk}}^{2}$$

If |r|≤1, the relationship between the roughness parameter and *r* is given by Figure 7, and the root mean square increases when |r| increases. If |r|<1, the skewness profile presents two local minima and two local maxima due to the competition of the [1,2], [2,0] and [2,2]-modes (Equation (Equation 15)), shown in Figure 7(b,b1). The kurtosis profile only presents one local maximum and two local minima for the same reason. The skewness and kurtosis profiles present rich complexity for r<0 compared with r>0. The opposite signs between the quadratic term and the quartic term result in a more complicated surface energy profile (the second and fourth quadrants in Figure 3), further creating various behaviours in the surface roughness parameters. Figure 7d presents the skewness-kurtosis plot, which is a prevalent tool in the application of lubrication [85,86], flows [87,88] and boundary layer analysis [89]. Figure 7d demonstrates a global binary plot with a single loop within −0.5<r<1, showing the complex coupling phenomenon of the two roughness parameters due to the lack of a dominant term for $\epsilon_{2}$ and $\epsilon_{4}$ in Equation (Equation 6) to Equation (Equation 8).

The maximum and minimum values of the skewness and kurtosis plots presented in Figure 8 within |r|≤2 are analyzed along with the computation of the kernel density (KD) of the surface profile at the corresponding *r*-value, which is defined by $\mathrm{KD}=\left({\bar{h}}_{\max}-{\bar{h}}_{\min}\right)\mathrm{PDF}\left(\bar{h}\right)/n$. Here, PDF is the probability distribution function, *n* denotes the number of bins for histogram, and n=1000 in this research for a smooth figure. The distribution KD of surface relief $\bar{h}$ is illustrated by the $\mathrm{KD}-\bar{h}$ profile in Figure 8. Figure 8a shows a right-leaning curve with a long-left tail (mean<median), which validates the fact that the surface profile under r=−2 has a negative skewness. Similarly, Figure 8b has a maximum, positive skewness, where the KD is a left-leaning curve. The skewness measures the distribution asymmetry, while kurtosis captures the sharpness of the distribution profile. Figure 8c suggests a more even KD profile than Figure 8d, which verifies that the surface under r=−0.454 has a smaller kurtosis than under r=+2. Figure 8 concludes again that the surface roughness features can be regulated by controlling the anchoring ratio *r* with a minimum kurtosis of Sku=2.13 (Figure 7). These results on roughness parameters can be further applied to tribology [67], topography [68] and analyzing shear mechanics [69].

## 4. Applications: Pathways to Targeted Surface Roughness Metrics

In practical applications, one would like to know the specific correlations between the most important surfaces roughness metrics and the anchoring coefficient ratio *r*. Figure 9 summarizes these key results in the parametric anchoring space $\left(\epsilon_{2},\epsilon_{4}\right)$.

Figure 9 summarizes the parametric sensitivity of the surface roughness metrics (root mean square, skewness, kurtosis) as a function of the anchoring coefficient ratio *r*, shown as straight lines (top) in the $\left(\epsilon_{2},\epsilon_{4}\right)$ plane. The *r*-ratio interval is from +2 to −2 (from the blue positive slope line to the red negative slope line). The constant *r*-lines intersect the horinzontal purple line $\epsilon_{4}=\mathrm{constant}$) at an ordered set of decreasing *r*-values, corresponding to the local extrema of the surface roughness metrics within the −2<r<+2 interval. As *r* decreases, the root mean square keeps declining until it reaches a minimum; then, it climbs up with increasing *r*. The skewness and kurtosis show a completely different profile. Within the range −2<r<+2, the skewness demonstrates two maximum and two minimum values, while the kurtosis presents one maximum and two minimum values. This phenomenon verifies that *r* plays a complex role in 2D surface wrinkling. Importantly, the local extrema in the surface roughness metrics occur at different values of *r*, and the presence of several extrema in the negative *r*-value interval shows that sector E1 of Figure 3 is a highly sensitive parametric area that can be exploited in future applications.

Figure 9 provides a road map to design egg carton surfaces with specific surface roughness metrics. For example, a self-assembled biocompatible cholesteric liquid crystal scaffold with the lowest root mean square can be produced while maintaining the anchoring coefficient ratio r≈−0.382. If we want to increase the surface shear strength, we need a surface with high positive skewness [69]. If *r* is negative, we can choose r≈−0.258, and if *r* is positive, the higher *r*, the better. The surface skewness and kurtosis are also shown to determine the contact angle and further influence the wettability of materials [90]. As mentioned above, the desired anchoring coefficients can be tuned by controlling the chemical composition and temperature [91].

## 5. Conclusions

Generating multifunctional wrinkled surfaces usually requires external stress loads, erosion and removal processes, or high-energy beams. In this paper, we present a platform to create nanowrinkling using an intrinsic surface pattern formation mechanism that exists in anisotropic chiral soft matter materials such as cholesteric liquid crystals. Here, shape is determined by internal capillary pressures generated by couplings and interactions between the geometry and the material structure. Specifically, the anisotropic interfacial energy depends on the coupling between the surface unit normal and the average molecular orientation of liquid crystals. Including chirality, spatial periodicity, and basic deformation modes generates additional capillary pressures not found in isotropic materials. The higher the coupling order between the geometry and the orientation, the more complex the surface energy landscape is and the greater the potential for complex surface wrinkling, such as egg carton surfaces, as demonstrated in this paper.

Specifically, in this paper, we studied the complex nanowrinkling patterns of cholesteric liquid crystals in a quartic anchoring model parameterized by two anchoring coefficients $\left(\epsilon_{2},\epsilon_{4}\right)$. For a given director field, the surface has 10 fundamental wrinkling modes. In a linear analysis of the quartic model (such that $\epsilon_{4}$ is never nonzero), the wrinkling morphology (the contribution of each fundamental wrinkling mode) is determined by the anchoring coefficient ratio $r=\epsilon_{2}/\left(2\epsilon_{4}\right)$, while the wrinkling magnitude is determined by the higher-order anchoring $\epsilon_{4}$ only. The surface wrinkling information is captured by a shape matrix, which reflects the specific role of the director-filed symmetry as well as the anchoring model. This efficient approach can easily be generalized to higher-order wrinkling by increasing the the dimensionality of the shape matrix as well as other director fields.

For functional surfaces that depend on specific surface roughness metrics, we can condense this information by performing a parametric study in terms of the anchoring coefficient ratio *r* and the parametric anchoring plane. Specifically, the higher-order moment information of surface geometry such as root mean square, skewness and kurtosis are evaluated, presenting complex nonlinear effects when the anchoring coefficient ratio is between −1 and +1. The linear analysis validates the analytic wrinkling profile as well as the scaling laws.

Taken together, this paper presents a theory and simulation platform for the design of nanowrinkled surfaces with targeted surface roughness metrics, generated by internal capillary pressures, of interest in the development of biomimetic multifunctional surfaces.

## Figures and Tables

**Figure 1 nanomaterials-12-01555-f001:**
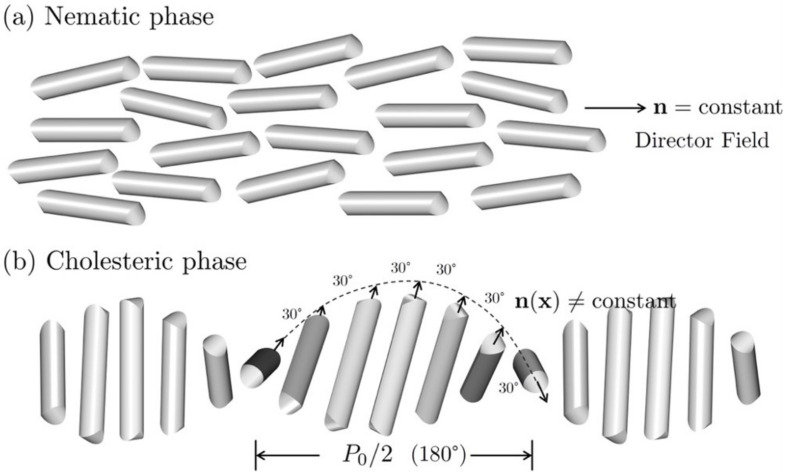
Nematic and cholesteric liquid crystal phases show different symmetry. (**a**) The nematic liquid crystal phase with an anisotropy axis n known as the director. (**b**) The chiral cholesteric liquid crystal phase with a spatial periodicity $P_{0}$.

**Figure 2 nanomaterials-12-01555-f002:**
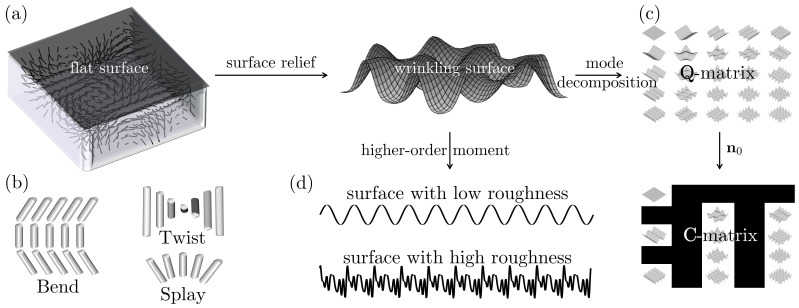
Schematic of this research. (**a**) A cholesteric liquid crystal tends to have a wrinkled surface rather than a flat surface driven by the director field (short lines) and the gradient of the director field. (**b**) The gradient of the director field shows three primary elastic modes: splay, bend and twist. (**c**) The wrinkling surface is the summation of various simple wrinkling modes, whose contributions are integrated into the C-matrix of fundamental wrinkling modes (shown in Equation (Equation 15)). (**d**) The higher-order moment of the wrinkling surface characterizes the surface roughness.

**Figure 3 nanomaterials-12-01555-f003:**
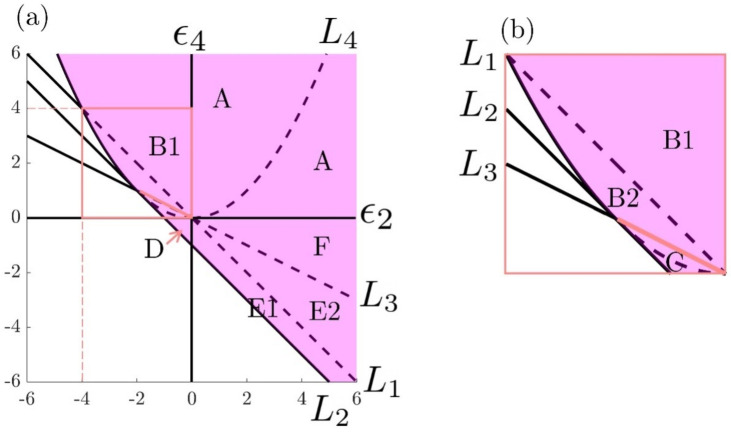
(**a**) Stability diagram of the Rapini–Papoular Equation (Equation 2) presented in the parametric anchoring energy space $\left(\epsilon_{2},\epsilon_{4}\right)$. (**b**) A zoom-in plot of the pink square ($-4<\epsilon_{2}<0$, $0<\epsilon_{4}<4$) in (**a**). $L_{1}$ to $L_{4}$ are critical curves separating different stable zones, where the local surface energy density shows various landscapes summarized and characterized in Table 1.

**Figure 4 nanomaterials-12-01555-f004:**
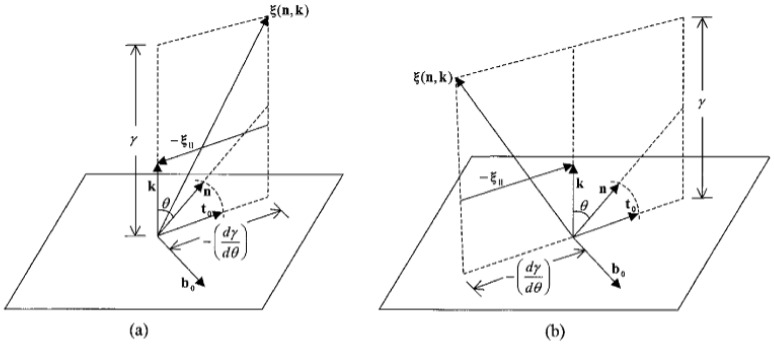
Schematic of the main vectors in the nematic Cahn-Hoffman vector thermodynamics for a (**a**) planar easy axis (−dγ/dθ>0) and (**b**) homeotropic easy axis (−dγ/dθ<0). The principal surface frame $\left(t_{0},b_{0}\right)$ is selected by the director orientation [80]. Reprinted (figure) with permission from Ae-Gyeong Cheong and Alejandro D. Rey, Physical Review E, 66, 021704 (2002). Copyright (2002) by the American Physical Society.

**Figure 5 nanomaterials-12-01555-f005:**
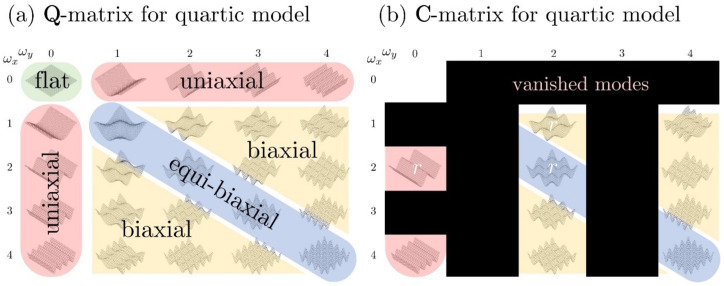
Visualization of the general shape Q- and C-matrix in Equations (Equation 13) and (Equation 15). The position of the wrinkling mode in Figure 5 exactly corresponds to the entry in the Q- and C-matrix. The red zone, blue zone, and yellow zone demonstrate uniaxial wrinkling, equibiaxial wrinkling and biaxial wrinkling, respectively. (**a**) The Q-matrix presents all possible wrinkling modes in a quartic model; (**b**) the C-matrix indicates that only 10 wrinkling modes are presented in this model due to the symmetry of the adopted director field.

**Figure 6 nanomaterials-12-01555-f006:**
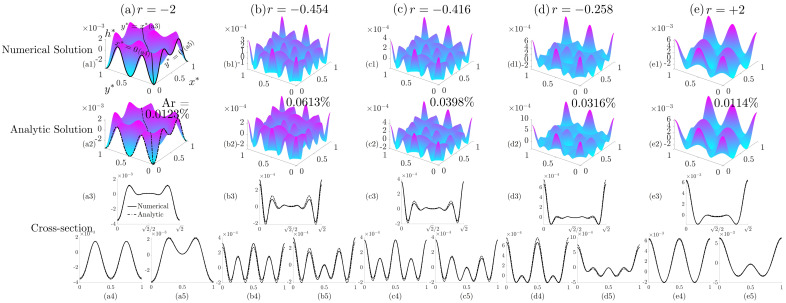
The numerical solution (first row, (**a1**)–(**e1**)), analytic solution (second row, (**a2**)–(**e2**)) and cross-section plots along a diagonal line $y^{*}=x^{*}$ ((**a3**)–(**e3**)) as well as two boundaries $x^{*}=0$ ((**a4**)–(**e4**)) and $y^{*}=0$ ((**a5**)–(**e5**)) with r=−2 (column (**a**), min skewness), r=−0.454 (column (**b**), min kurtosis), r=−0.416 (column (**c**), zero skewness), r=−0.258 (column (**d**), max skewness) and r=+2 (column (**e**), max kurtosis) by fixing $\epsilon_{4}=-0.01$. The percentage values shown in the second row are the average difference ratio Ar defined in Equation (Equation 16). The full and dashed-line curves for the cross-sectional plots represent numerical solution and analytic solutions, respectively.

**Figure 7 nanomaterials-12-01555-f007:**
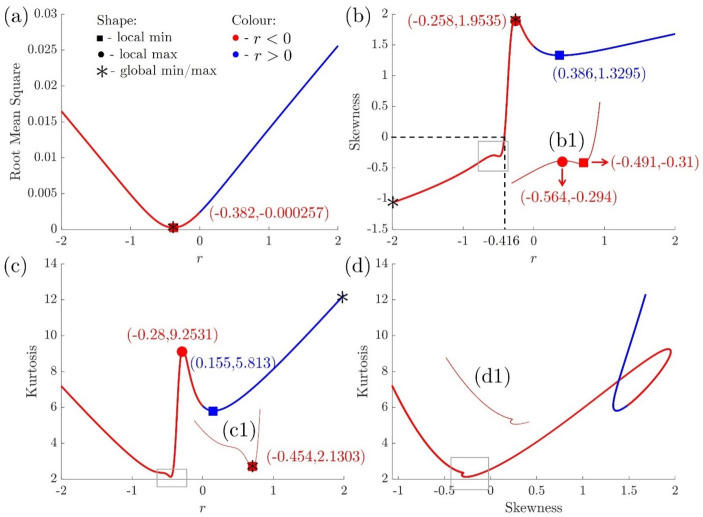
Surface roughness parameters of $\bar{h}$ are dependent on the anchoring coefficient ratio *r*. (**a**) Root mean square (${\mathrm{Sq}}^{2}$), (**b**) skewness (Ssk) and (**c**) kurtosis (Sku) are functions of *r*. (**d**) Sku as a function of Ssk parametrized by *r*. The red and blue curves demonstrate r<0 and r>0, respectively. A square marker represents a local minimum, and a filled circle represents a local maximum. A star marker indicates a global extremum. (**b1**,**c1**,**d1**) are the zoom-in details of the region with squares in (**b**–**d**), respectively.

**Figure 8 nanomaterials-12-01555-f008:**
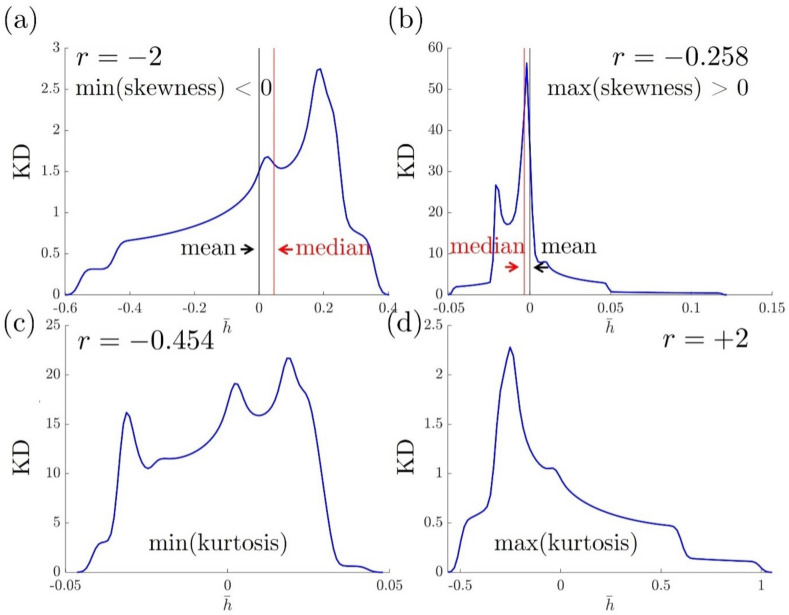
The kernel density (KD) when (**a**) r=−2 (minimum skewness), (**b**) r=−0.258 (maximum skewness), (**c**) r=−0.454 (minimum kurtosis) and (**d**) r=+2 (maximum kurtosis).

**Figure 9 nanomaterials-12-01555-f009:**
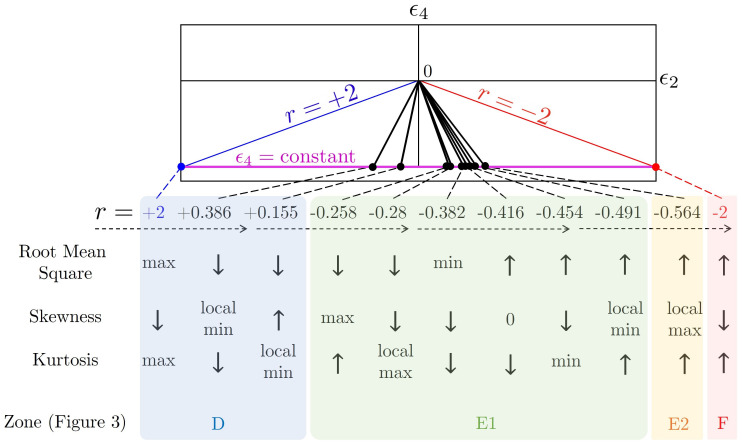
Summary of the parametric sensitivity of the key surface roughness metrics: root mean square, skewness, and kurtosis (Equations (Equation 17) and (Equation 18)). A straight line with slope 1/(2r) passing through the origin in the parametric space $\left(\epsilon_{2},\epsilon_{4}\right)$ corresponds to a morphology (evaluated by C(r)-matrix), and its *y*-axis coordinate (purple line $\epsilon_{4}=\mathrm{constant}$) determines the amplitude of the wrinkling profile. The location of these *r*-points in parametric space (Figure 3) are: r=−2, zone F; r=−0.564, zone E2; r=−0.491 to −0.258 are in zone E1; and r=+0.155 to +2 are in zone D. The density of the special surface roughness metrics in the negative *r*-value interval indicates that zone E1 in Figure 3 is a highly sensitive area.

**Table 1 nanomaterials-12-01555-t001:** A summary of local energy density extrema in the sub-regions presented in Figure 3.

Regions in Figure 3	Local Minimum	Local Maximum
A	n⊥k	n‖k
B1	${\left(n\;\cdot \;k\right)}^{2}=\left|r\right|$	n‖k
B2	${\left(n\;\cdot \;k\right)}^{2}=\left|r\right|$	n⊥k
C	n‖k	n⊥k
D	n‖k	n⊥k
E1	n⊥k	${\left(n\;\cdot \;k\right)}^{2}=\left|r\right|$
E2	n‖k	${\left(n\;\cdot \;k\right)}^{2}=\left|r\right|$
F	n⊥k	n‖k

## Data Availability

Not applicable.

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
