# Peer review of "Complex Nanowrinkling in Chiral Liquid Crystal Surfaces: From Shaping Mechanisms to Geometric Statistics"

_nanomaterials, 2022, doi:10.3390/nano12091555_

Round 1
Reviewer 1 Report
In this manuscript, Ziheng Wang and coworkers present complex nanowrinkling in chiral liquid crystal surfaces. It was shown that such an approach allows to develop a theory and simulation platform for the design of nano-wrinkled surfaces with targeted surface roughness metrics, generated by internal capillary pressures, of interest in the development
of biomimetic multifunctional surfaces.
This manuscript contains new prospective ideas in the fabrication of nanostructured surfaces that will be of interest to the readers of Nanomaterials mdpi after minor revision. There are several concerns the authors need to address. These are listed below:
- A strong limitation of the manuscript is lack of the experimental studies. I am not sure that this model will work.
- What type of liquid crystals should be used for this model. The chemistry of the liquid crystal is very different even for nematics.
- In turn to the previous question, I didn't find information about the temperature regime of the systems. In general, we know that liquid crystals are dynamic systems. What about surface geometry in dynamic systems, it is not clear to me.
- I would like to recommend to cite some important papers that will allow to improve quality of the paper:
https://doi.org/10.1021/acs.langmuir.6b02946
https://doi.org/10.1016/j.apsusc.2017.03.001 (where roughness parameters of the grafted liquid crystal brushes were controlled by temperature)
https://hal.mines-ales.fr/hal-03247385
Reviewer 2 Report
In this manuscript, the authors present a theoretical study and develop a simulation platform for the design of nano-wrinkled surfaces at the interface between nematic liquid crystals and isotropic media. This method could have impact on the development of biomimetic multifunctional surfaces.
In the present context, the wrinkled surfaces originate from the coupling between the surface geometry and the surface molecular orientation: for orientation scales of the order of micrometers the surface wrinkling scale is of the order of nanometers.
The developed model is based on the liquid crystal shape equation and the intrinsic capillary pressures – related to splay, twist and bend deformations – are the only internal interactions accounted for. Morevover, the focus of the model is restricted to equilibrium patterns and all time-dependent phenomena have been neglected. The authors illustrate how different modes impact final shapes explicitly, by founding optimal ways to synthesize information for the wrinkling patterns using a shape classification based on the superposition of primary modes.
As a general remark, the paper content is interesting, and the manuscript is well organized. However, exposition and grammar need to be cleaned to help readers grasp the meaning of this work in depth.
Some of the mistakes/issues I noticed are listed in the following:
- Line 69, “this simplest model”;
- Line 77, between “If r is relatively small” and “material orientation scales …”, “,” is required
- Line 79, “,” after n(x) should be cancelled out;
- Lines 100-101, “The statistical properties …” ARE “described”;
- Lines 117-118, “bend, twist and bend \nabla n”, bend is used twice and splay should be included; \nabla n must be cancelled out;
- Line 116, what do the authors mean by “high order interfacial energy formulation”? Clarify, please.
- Line 176, \gamma is a function of (n.k)^2 and not simply (n.k). I would explain this depends on the equivalence between n and -n in non-polar liquid crystals.
- The parameter r is dubbed as anchoring coefficient ratio on line 176 (please replace “ration” with “ratio”). Please use this name consistently all over the manuscript.
- Lines 193-194, the sentence is totally unclear
- Lines 195-196, please fix “… tensors. without loss …”
- In Eq. (9), please fix the gradient over the surface \sigma of the molecular director.
